# Large language models streamline automated machine learning for clinical studies

Soroosh Tayebi Arasteh [1] ✉, Tianyu Han [1] ✉, Mahshad Lotfinia [1,2], Christiane Kuhl[1], Jakob Nikolas Kather [3,4], Daniel Truhn [1,5] & Sven Nebelung [1,5]

A knowledge gap persists between machine learning (ML) developers (e.g., data scientists) and practitioners (e.g., clinicians), hampering the full utilization of ML for clinical data analysis. We investigated the potential of the ChatGPT Advanced Data Analysis (ADA), an extension of GPT-4, to bridge this gap and perform ML analyses efficiently. Real-world clinical datasets and study details from large trials across various medical specialties were presented to ChatGPT ADA without specific guidance. ChatGPT ADA autonomously developed state-of-the-art ML models based on the original study's training data to predict clinical outcomes such as cancer development, cancer progression, disease complications, or biomarkers such as pathogenic gene sequences. Following the re-implementation and optimization of the published models, the head-to-head comparison of the ChatGPT ADA-crafted ML models and their respective manually crafted counterparts revealed no significant differences in traditional performance metrics ($p \geq 0.072$). Strikingly, the ChatGPT ADA-crafted ML models often outperformed their counterparts. In conclusion, ChatGPT ADA offers a promising avenue to democratize ML in medicine by simplifying complex data analyses, yet should enhance, not replace, specialized training and resources, to promote broader applications in medical research and practice.

Machine learning (ML) drives advancements in artificial intelligence and is about to transform medical research and practice, especially in diagnosis and outcome prediction[1,2]. Recently, the adoption of ML for analyzing clinical data has expanded rapidly. Today, ML models have an established and evolving role in various areas of public health and medicine, spanning image analysis, public health, clinical-trial performance, and operational organization[2]. ML models are used in variable contexts such as augmenting medical knowledge, assisting clinicians, or taking on administrative tasks[3]. Several developments, such as increases in (i) available data generated during clinical care, (ii) available computational processing capacities, and (iii) research activities, favor the more widespread future utilization of ML models in medicine[4]. However, the complexity of developing, implementing, and validating those models renders them inaccessible to most clinicians and medical researchers[5]. It also limits their utilization to those people or groups that combine expertise in medicine and data science.

Automated machine learning (AutoML) is an established discipline that aims to make ML accessible to non-technical experts. In

[1]Department of Diagnostic and Interventional Radiology, University Hospital RWTH Aachen, Aachen, Germany. [2]Institute of Heat and Mass Transfer, RWTH Aachen University, Aachen, Germany. [3]Else Kroener Fresenius Center for Digital Health, Medical Faculty Carl Gustav Carus, Technical University Dresden, Dresden, Germany. [4]Medical Oncology, National Center for Tumor Diseases (NCT), University Hospital Heidelberg, Heidelberg, Germany. [5]These authors jointly supervised this work: Daniel Truhn, Sven Nebelung. ✉e-mail: soroosh.arasteh@rwth-aachen.de; than@ukaachen.de

medicine, the principle feasibility and use of AutoML platforms, such as the Classification Learner of MATLAB (MathWorks Inc.), Vertex AI (Google LLC), and Azure (Microsoft Corporation), have been demonstrated[6–11], enabling non-technical experts to create ML models. These software solutions automate algorithm training and fine-tuning by providing dedicated interfaces to build and run a particular ML model. The user needs to direct the software to the desired output. So far, however, models using natural language commands and their conversion to Python code have not been implemented.

Powerful large language models (LLMs)[12], such as ChatGPT's latest version, GPT-4[13] (Generative Pre-Trained Transformer-4, OpenAI, CA, US), expand the repertoire of AutoML platforms by offering a well-accessible option to the user[14,15]. While conversing with humans in plain language, LLMs can reason and perform logical deduction. Recently, the ChatGPT Advanced Data Analysis (ADA), formerly known as ChatGPT Code Interpreter, has been made available as an extension and beta feature that may be used to analyze data and math problems, create charts, and write, execute, and refine computer code[16]. Instructing ChatGPT ADA can be straightforward, such as "Analyze this patient data and build a machine-learning model predicting 12-month mortality rates". Given this prompt, ChatGPT ADA will execute the task and provide feedback on the procedure. However, its validity and reliability in advanced data processing and analysis for large clinical trials have not yet been evaluated.

Our objective was to study the validity and reliability of ChatGPT ADA in autonomously developing and implementing ML methods. We included real-world datasets from four large clinical trials of various medical specialties that applied ML models for advanced data analysis (Fig. 1). We hypothesized that (i) ChatGPT ADA may be used intuitively and does not require prior training, resources, and guidance in ML theory and practice to implement advanced ML methods efficiently and accurately and that (ii) the results of these implementations match those of specialized data scientists. We provide evidence that advanced LLMs like ChatGPT ADA simplify complex ML methods, increasing their accessibility in medicine and beyond.

## Results

Across four large clinical-trial datasets, ChatGPT ADA autonomously formulated and executed advanced ML techniques for disease screening and prediction. Its performance matched the hand-crafted and customized ML methods re-implemented based on the original studies. Figure 2 illustrates an exemplary interaction with ChatGPT ADA, highlighting the prompts and responses for autonomous prediction.

After briefly summarizing each clinical trial and associated dataset, we compare the ML methods head-to-head for each trial. We include ML methods developed and executed by ChatGPT ADA against the performance metrics of the originally published ML methods (as reported in the original studies) and the validatory ML methods (as re-implemented by a seasoned data scientist, S.T.A. with five years of experience in ML). Because individual patient predictions were unavailable in the original studies, the best-performing ML methods of the original studies were re-implemented. We conclude our analysis by

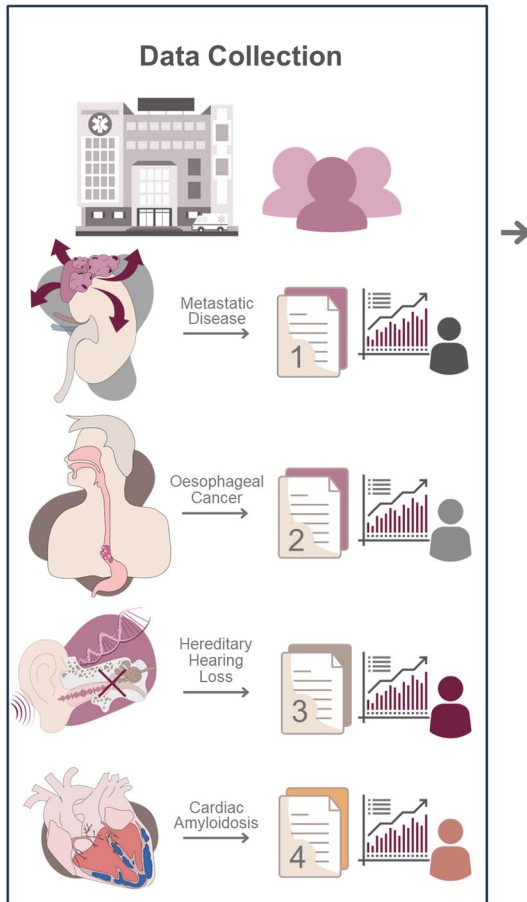
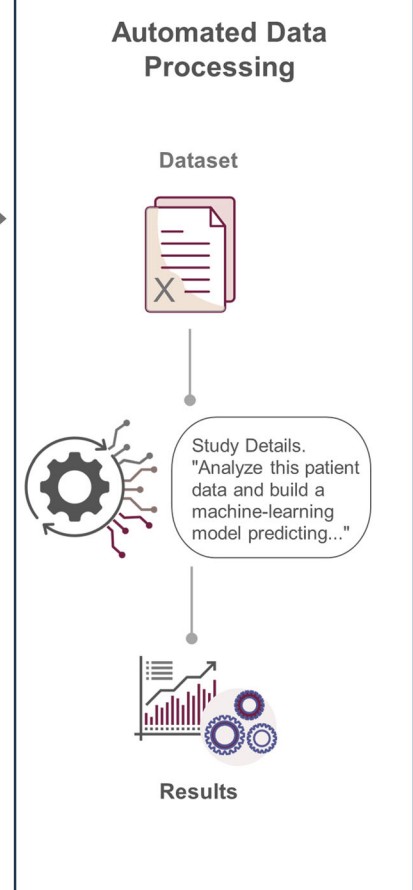
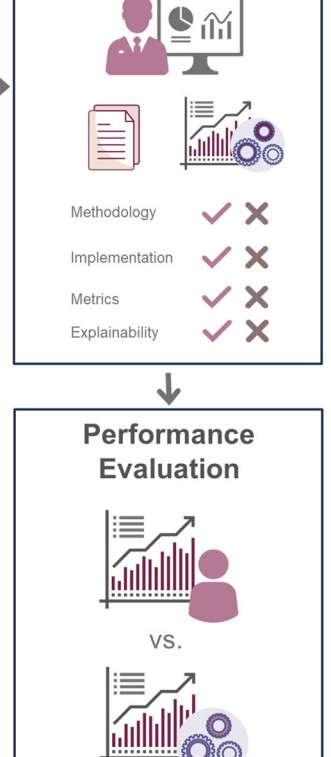

**Fig. 1 | Study design.** Real-world datasets and study details from four large clinical trials were collected and input into the ChatGPT Advanced Data Analysis (ADA) tool. The tool autonomously selected the appropriate machine-learning models for the analysis following prompting. The models were expert-checked and comprehensively evaluated. The ChatGPT ADA-based predictions were compared to the original studies (benchmark publication) and the validatory predictions following the re-implementation of the models. Figure 1 was provided by a freelancer service (fiverr.com). Copyright rests with the authors. The figure constitutes original material and has not been published before.

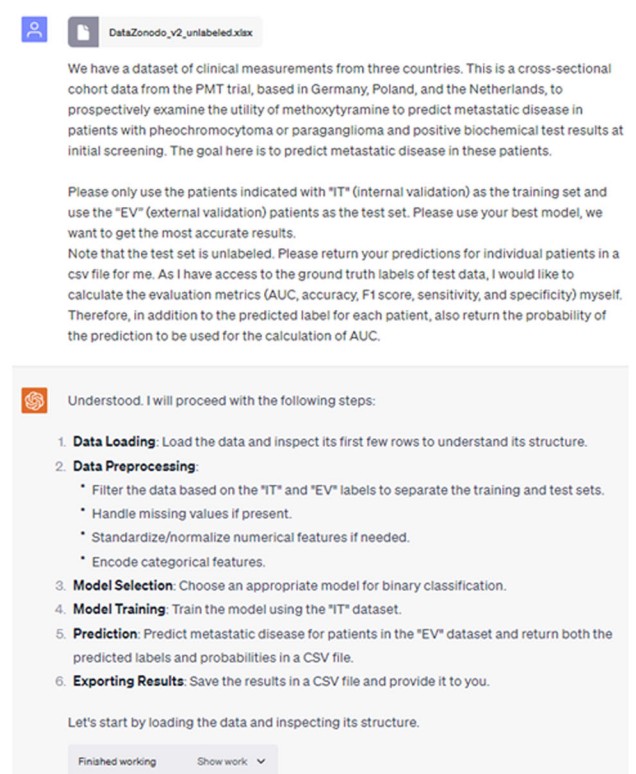

**Fig. 2 | Screenshots of an example interaction with ChatGPT ADA to analyze the endocrinologic oncology dataset.** ChatGPT ADA autonomously selects and applies the appropriate ML model for the provided dataset, generating predictions for the test data. The model also displays deeper insights in response to follow-up queries about the reasoning and parameters guiding its choices. Note: The "Show work" option visible in the images allows users to view the intermediary Python code offered by the tool.

presenting the explainability metrics determined by ChatGPT ADA and confirmed by our re-implementation.

## Metastatic disease [endocrinologic oncology]–predicting metastatic disease in pheochromocytoma and paraganglioma

Pamporaki et al. utilized cross-sectional cohort data from the US, Germany, Poland, and the Netherlands, and employed ML methods to predict metastatic disease in patients diagnosed with pheochromocytoma or paraganglioma using blood test results[17]. These tumors are referred to as the 'great masquerader' because of their unspecific clinical presentation secondary to largely variable catecholamine excess, which poses diagnostic challenges[17]. The original study's training and test set cohorts comprised 493 and 295 patients (Table 1). Using predictions by 12 clinical experts as their reference, the authors implemented multiple supervised ML models, i.e., the decision tree classifier, support vector machine, Naïve Bayes, and AdaBoost[18] ensemble tree classifier. In the original study, the latter model performed best and significantly outperformed the clinical care specialists, with an area under the receiver operating characteristic curve (AUROC) of 0.942 versus 0.815 (best-performing clinical expert, $p < 0.001$). Using the same (training and test) dataset distribution as the original study but withholding specific guidance on data preprocessing or ML methodology, we prompted ChatGPT ADA to predict metastatic disease in the test set while. ChatGPT ADA selected a Gradient Boosting Machine (GBM)[19] model for its prediction and achieved a slightly improved performance relative to its best-performing published counterpart in terms of AUROC values (0.949 vs. 0.942), accuracy (0.922 vs. 0.907), and F1-scores (0.806 vs. 0.755) (Table 2). The entire conversation with ChatGPT ADA regarding prompts and responses is detailed in Supplementary Note 1.

After re-implementing and optimizing the best-performing ML model from the original study, i.e., the AdaBoost[18] ensemble tree classifier, as our validatory ML model, we performed a head-to-head comparison. The performance metrics were similar (validatory re-implementation: AUROC = 0.951 ± 0.014 [95% CI: 0.920, 0.977]; ChatGPT ADA: AUROC = 0.949 ± 0.015 [95% CI: 0.917, 0.974]) and not significantly different ($p = 0.464$) (Table 3 and Fig. 3).

## Esophageal cancer [gastrointestinal oncology]–predicting carcinoma of the distal esophagus and oesophagogastric junction

Gao et al. used sponge cytology testing and epidemiologic data to screen for esophageal squamous cell carcinoma and adenocarcinoma of the oesophagogastric junction[20]. The authors obtained multicohort data from 14,597 participants in China (Table 1) to design six ML models, i.e., logistic regression, adaptive boosting, Light Gradient Boosting Machine (LightGBM)[21], extreme gradient boosting, Random Forest (RF)[22], and support vector machine[23], to predict high-grade intraepithelial neoplasia and carcinoma based on 105 cytologic and 15 epidemiologic features. The best-performing model was the LightGBM, which achieved an AUROC value of 0.960 in the test set. In contrast, ChatGPT ADA selected the GBM and outperformed the original model at an AUROC value of 0.979 (Table 2). Supplementary Note 2 details the entire conversation with ChatGPT ADA for this dataset.

The head-to-head analysis of the ChatGPT ADA-selected ML model and our validatory re-implemented ML model indicated largely similar AUROC values of 0.979 ± 0.004 [95% CI: 0.970, 0.986] and 0.978 ± 0.005 [95% CI: 0.967, 0.986], respectively, which were not significantly different ($p = 0.496$) (Table 3 and Fig. 3).

**Table 1 | Characteristics of the clinical trials whose datasets were included**

| | Metastatic disease[17] [endocrinologic oncology] | | Esophageal cancer[20] [gastrointestinal oncology] | | Hereditary hearing loss[24] [otolaryngology] | | Cardiac Amyloidosis[25] [cardiology] (*) | |
|---|---|---|---|---|---|---|---|---|
| | Training set | Test set | Training set | Test set | Training set | Test set | Training set | Test set |
| Patient number | | | | | | | | |
| Total [n] (with disease/without disease [%]) | 493 (34/66) | 295 (19/81) | 7899 (3/97) | 6698 (2/98) | 1209 (76/24) | 569 (77/23) | 1712 (50/50) | 430 (50/50) |
| Patient sex | | | | | | | | |
| Female/male [%] | 49/51 | 57/43 | 0/100 | 0/100 | 49/51 | 39/61 | N/A | N/A |
| Patient age [years] | | | | | | | | |
| Median Mean ± standard deviation Range (minimum, maximum) | 42 42 ± 18 (4, 83) | 48 47 ± 16 (11, 82) | 56 56 ± 9 (39, 82) | 55 56 ± 9 (24, 86) | N/A 18 ± 15 (N/A, N/A) | N/A 34 ± 12 (N/A, N/A) | N/A | N/A |
| Location of clinical trial | US, Netherlands | Germany, Poland, Netherlands | China | China | China | China | US | US |

(*) indicates that the original data split and, consequently, the external validation dataset was unavailable per the original study. In line with the published methodology, we randomly allocated 80% of patients and controls to the training set (n = 1712) and 20% to the test set (n = 430). N/A not available.

**Table 2 | Benchmark publication—ML models and their published performance metrics as a function of clinical-trial dataset**

| | AUROC | Accuracy | F1-score | Sensitivity | Specificity |
|---|---|---|---|---|---|
| Metastatic disease [endocrinologic oncology][17] | | | | | |
| Best-performing ML model (original study): AdaBoost[18] ensemble tree | 0.942 | 0.907 | 0.755 | 0.833 | 0.922 |
| ChatGPT ADA: GBM | 0.949 | 0.922 | 0.806 | 0.841 | 0.941 |
| Best-performing clinical expert | 0.815 | 0.830 | N/A | 0.800 | 0.850 |
| Mean of clinical experts [n = 12] | 0.710 | 0.722 | N/A | 0.664 | 0.755 |
| Esophageal cancer [gastrointestinal oncology][20] | | | | | |
| Best-performing ML models (original study): LightGBM | 0.960 | N/A | N/A | 0.945 | 0.919 |
| ChatGPT ADA: GBM | 0.979 | 0.985 | 0.538 | 0.457 | 0.995 |
| Hereditary hearing loss [otolaryngology][24] | | | | | |
| Best-performing ML model (original study): Support vector machine | 0.751 | 0.812 | 0.861 | 0.925 | N/A |
| ChatGPT ADA: RF | 0.773 | 0.767 | 0.845 | 0.834 | 0.541 |
| Mean of clinical experts [n = 3] | N/A | N/A | N/A | 0.789 | 0.470 |
| Cardiac amyloidosis [cardiology][25] | | | | | |
| Best-performing ML model (original study): RF | 0.930 | 0.870 | 0.875 | 0.870 | 0.870 |
| ChatGPT ADA: RF | 0.954 | 0.892 | 0.894 | 0.903 | 0.884 |

Indicated are the performance metrics of the best-performing ML models as published in the original studies, of the ChatGPT ADA-based ML models, and, if available, of individual or numerous clinical experts.

AdaBoost adaptive boosting, AUROC area under the receiver operating characteristic curve, ChatGPT ADA ChatGPT advanced data analysis, GBM gradient boosting machine, LightGBM light gradient boosting machine, N/A not available, RF random forest.

## Hereditary hearing loss [otolaryngology]−predicting pathogenic genetic variants

Luo et al. aimed to identify patients with hereditary hearing loss based on particular gene sequences, i.e., the sequence variants at 144 sites in three genes[24]. Using data from 1778 patients and controls (Table 1), the authors implemented six supervised ML models, i.e., the decision tree, random forest, k-nearest neighbor, adaptive boosting, multilayer perceptron models, and the support vector machine[23]. The latter ML method performed best (AUROC value of 0.751) and outperformed three clinical experts. The ChatGPT ADA-selected predictive model, i.e., the RF classifier, outperformed the original model regarding AUROC values (0.773) yet demonstrated inferior performance regarding accuracy (0.767 vs. 0.812) and F1-score (0.845 vs. 0.861) (Table 2). Supplementary Note 3 details the entire conversation with ChatGPT ADA for this dataset.

The head-to-head analysis of the ChatGPT ADA-selected ML model and our validatory re-implemented ML model indicated largely similar AUROC values of 0.773 ± 0.024 [95% CI: 0.726, 0.817] and

0.762 ± 0.026 [95% CI: 0.714, 0.812], respectively, which were not significantly different (p = 0.624) (Table 3 and Fig. 3).

## Cardiac amyloidosis [cardiology]−predicting the cardiomyopathies

Huda et al. attempted to identify patients at risk of cardiac amyloidosis, a now treatable condition predisposing to heart failure, using various cohorts and established medical diagnoses retrieved from health records[25]. Using data from 2142 patients and controls (Table 1), the authors designed three ML models, i.e., logistic regression, extreme gradient boosting, and RF classifier. They found the latter ML model to perform best (AUROC value of 0.930 [internal validation set]). Because the external validation dataset was not publicly available, we used the original study's internal validation set to prompt ChatGPT ADA as above. The ChatGPT ADA-selected predictive model, i.e., the RF classifier, outperformed the original model regarding the AUROC (0.954) and the other performance metrics (Table 2). Supplementary Note 4 details the entire conversation with ChatGPT ADA for this dataset.

**Table 3 | Benchmark validatory re-implementation—ML models and their performance metrics as a function of clinical-trial dataset**

| | AUROC | Accuracy | F1-score | Sensitivity | Specificity |
|---|---|---|---|---|---|
| **Metastatic disease [endocrinologic oncology][17]** | | | | | |
| Validation model (AdaBoost ensemble tree) | 0.951 ± 0.014 [0.920, 0.977] | 0.911 ± 0.016 [0.878, 0.942] | 0.783 ± 0.041 [0.698, 0.915] | 0.821 ± 0.050 [0.720, 0.915] | 0.932 ± 0.016 [0.900, 0.962] |
| ChatGPT ADA (GBM) | 0.949 ± 0.015 [0.917, 0.974] | 0.922 ± 0.016 [0.892, 0.953] | 0.806 ± 0.039 [0.727, 0.876] | 0.841 ± 0.050 [0.742, 0.933] | 0.941 ± 0.015 [0.909, 0.969] |
| p-value | 0.464 | 0.665 | 0.659 | 0.619 | 0.646 |
| **Esophageal cancer [gastrointestinal oncology][20]** | | | | | |
| Validation model (LightGBM) | 0.978 ± 0.005 [0.967, 0.986] | 0.986 ± 0.001 [0.983, 0.989] | 0.576 ± 0.041 [0.492, 0.652] | 0.497 ± 0.045 [0.411, 0.585] | 0.996 ± 0.001 [0.994, 0.997] |
| ChatGPT ADA (GBM) | 0.979 ± 0.004 [0.970, 0.986] | 0.985 ± 0.001 [0.982, 0.988] | 0.538 ± 0.043 [0.452, 0.620] | 0.457 ± 0.044 [0.370, 0.541] | 0.995 ± 0.001 [0.994, 0.997] |
| p-value | 0.496 | 0.271 | 0.267 | 0.238 | 0.404 |
| **Hereditary hearing loss [otolaryngology][24]** | | | | | |
| Validation model (support vector machine) | 0.762 ± 0.026 [0.714, 0.812] | 0.783 ± 0.018 [0.747, 0.817] | 0.860 ± 0.012 [0.836, 0.884] | 0.869 ± 0.016 [0.836, 0.899] | 0.503 ± 0.043 [0.419, 0.584] |
| ChatGPT ADA (RF) | 0.773 ± 0.024 [0.726, 0.817] | 0.767 ± 0.018 [0.733, 0.800] | 0.845 ± 0.013 [0.820, 0.869] | 0.834 ± 0.018 [0.795, 0.867] | 0.541 ± 0.044 [0.453, 0.628] |
| p-value | 0.624 | 0.249 | 0.198 | 0.072 | 0.741 |
| **Cardiac amyloidosis [Cardiology][25]** | | | | | |
| Validation model (RF) | 0.952 ± 0.010 [0.931, 0.969] | 0.890 ± 0.015 [0.858, 0.919] | 0.892 ± 0.016 [0.860, 0.920] | 0.893 ± 0.020 [0.853, 0.932] | 0.888 ± 0.021 [0.847, 0.928] |
| ChatGPT ADA (RF) | 0.954 ± 0.010 [0.934, 0.972] | 0.892 ± 0.015 [0.863, 0.921] | 0.894 ± 0.016 [0.862, 0.922] | 0.903 ± 0.020 [0.860, 0.938] | 0.884 ± 0.023 [0.841, 0.926] |
| p-value | 0.539 | 0.525 | 0.539 | 0.647 | 0.460 |

Indicated are the performance metrics of the re-implemented and optimized ML models (as reported to perform best in the original studies) and of the ChatGPT ADA-based ML models. A seasoned data scientist re-implemented the ML models for validation purposes, thereby making per-patient predictions and head-to-head comparisons using bootstrapping. Performance metrics are presented as mean ± standard deviation [95% Confidence Intervals]. Bootstrapping[54] with replacements and 1000 redraws on the test sets (number of independent samples: endocrinologic oncology dataset, n = 295; Gastrointestinal Oncology dataset, n = 6698, Otolaryngology dataset, n = 569; Cardiology dataset, n = 430) was applied to determine means and measures of statistical spread in terms of standard deviations and 95% confidence intervals and if the metrics were significantly different if the metrics were significantly different. We adjusted for multiple comparisons based on the false discovery rate, setting the family-wise alpha threshold at 0.05. Source data are provided as a Source Data file.
*AdaBoost* adaptive boosting, *AUROC* area under the receiver operating characteristic curve, *ChatGPT ADA* ChatGPT advanced data analysis, *GBM* gradient boosting machine, *LightGBM* light gradient boosting machine, *RF* random forest.

The head-to-head analysis of the ChatGPT ADA-selected ML model and our validatory re-implemented ML model indicated largely similar AUROC values of $0.954 \pm 0.010$ [95% CI: 0.934, 0.972] and $0.952 \pm 0.010$ [95% CI: 0.931, 0.969], respectively, which were not significantly different ($p = 0.539$) (Table 3 and Fig. 3).

## Explainability analysis

The interpretation of model predictions, especially in situations demanding transparency and trust, relies on our capacity to grasp the importance of individual features. To study the ability of ChatGPT to provide metrics of explainability, we utilized the SHapley Additive exPlanations (SHAP)[26] analysis that helps quantify each feature's contributions to a model's predictions. We instructed ChatGPT ADA to perform the SHAP analysis autonomously without providing specific guidance. Figure 4 details the top 10 most influential features (ranked by their overall impact as determined by the mean absolute SHAP values) contributing to the best-performing ML model of each clinical trial. SHAP values measure a feature's influence on a model's output. High absolute SHAP values signify substantial impact, and positive SHAP values elevate the model's prediction above the baseline.

## Discussion

The availability of LLMs for advanced data processing[27,28], specifically those with the capacity to write, execute, and refine code like ChatGPT ADA, marks a pivotal shift in the convergence of data science and clinical research and practice. Our investigation of four large clinical trials underscores the potential of these tools to simplify complex ML methods and increase their accessibility in medicine and beyond. If implemented with due diligence, these tools enhance, not replace, specialized training and resources, democratizing access to advanced data processing and, potentially, revolutionizing data-driven medicine.

While ML and "Big Data" are touted as revolutionizing healthcare[29], clinicians regularly deal with too many patients in too little time[30]. Yet, they make hundreds of decisions each day that are primarily based on eminence and not on published data or studies[31]. Consequently, a valid and reliable tool that automates data processing may decentralize the monopoly of evidence held by specialized institutions. While clinicians remain at the center of patient care, ML methods can assist their expertise, e.g., by identifying at-risk patients for specific conditions based on electronic health records or by analyzing complex datasets such as genomic sequences. Intentionally, we

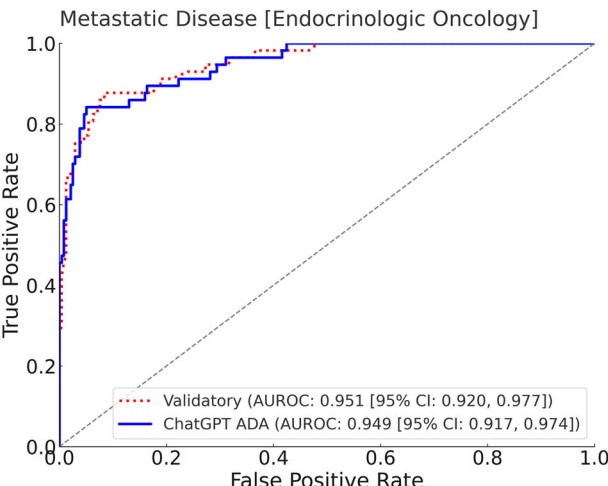

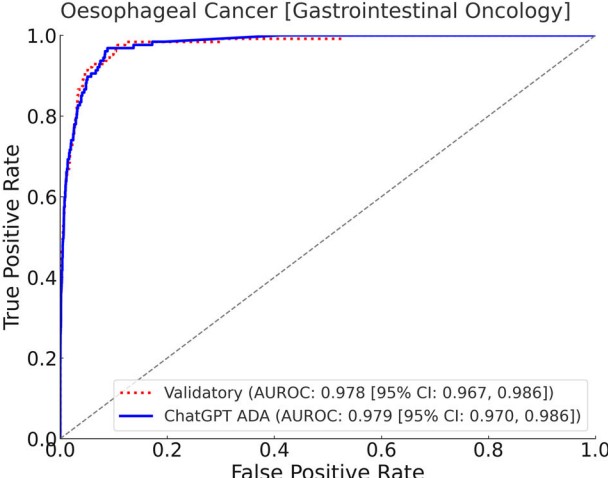

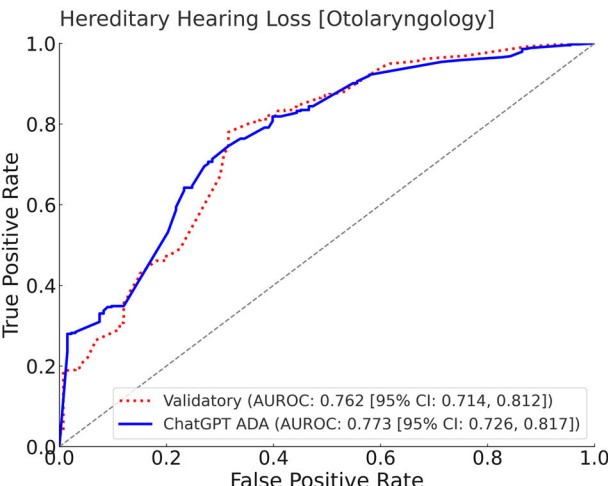

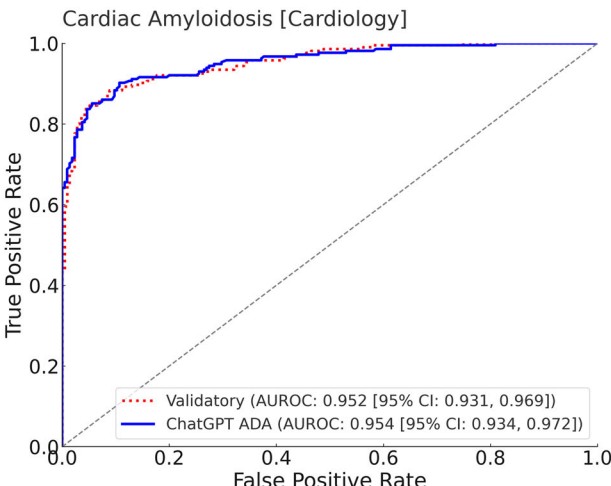

**Fig. 3 | Benchmark validatory re-implementation–receiver operating characteristic (ROC) curves of ML models as a function of the clinical-trial dataset.** The ROC curves of the ChatGPT ADA-based ML model (blue, solid curve) and the validatory ML model as re-implemented by a seasoned data scientist (red, dotted curve) are shown. The True Positive Rate (sensitivity) is plotted versus the False Positive Rate (1-specificity). The diagonal gray line represents the line of no discrimination. Source data are provided as a Source Data file. Bootstrapping[54] with replacements and 1000 redraws on the test sets (number of independent samples: Endocrinologic Oncology dataset, $n = 295$; Gastrointestinal Oncology dataset, $n = 6698$, Otolaryngology dataset, $n = 569$; Cardiology dataset, $n = 430$) was applied to determine means and measures of statistical spread, i.e., standard deviations and 95% confidence intervals (CI). AUROC area under the receiver operating characteristic curve, ChatGPT advanced data analysis.

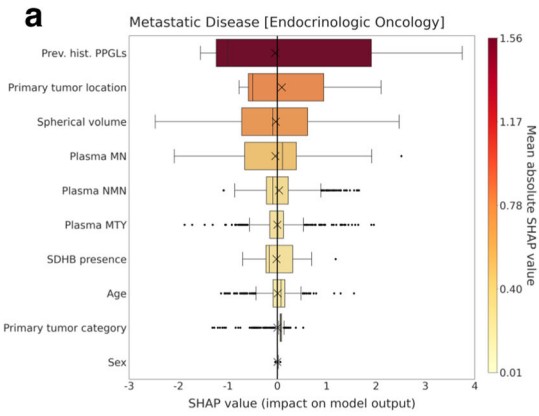

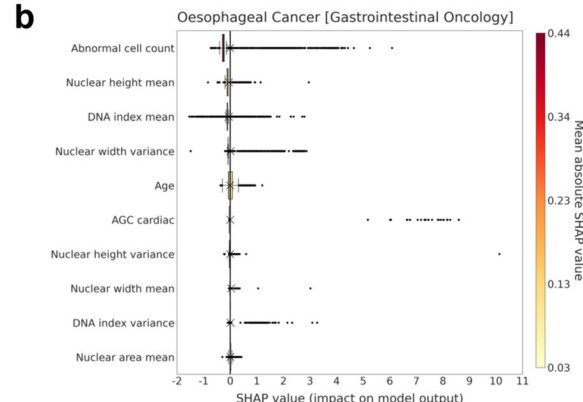

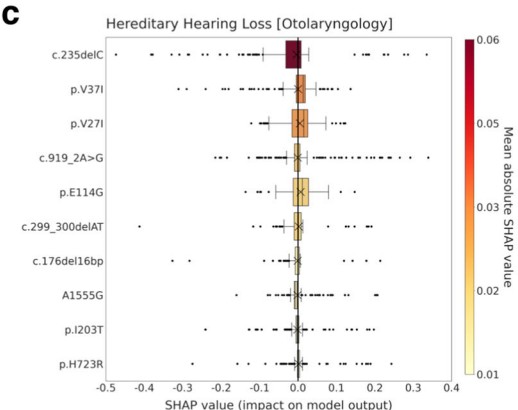

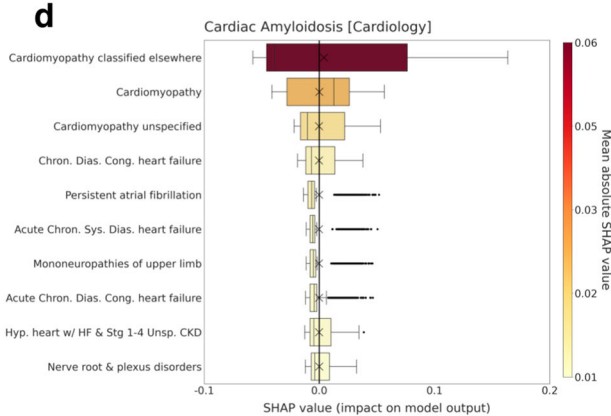

**Fig. 4 | Model explainability through the top 10 predictive features for the ChatGPT ADA-selected machine-learning models.** An explainability analysis was performed for each clinical trial including (**a**) Metastatic Disease [Endocrinologic Oncology], **b** Oesophageal Cancer [Gastrointestinal Oncology], **c** Hereditary Hearing Loss [Otolaryngology], and **d** Cardiac Amyloidosis [Cardiology], and ChatGPT ADA-selected machine-learning model. Indicated are SHapley Additive exPlanations (SHAP) values of each predictive feature that measure the feature's influence on model predictions. High absolute SHAP values signify substantial influence. The features are ranked from top to bottom based on the mean absolute SHAP values (color-coded on the right). In **c**, specific gene locations are indicated. Please refer to the Methods for more details on abbreviations. Box plots indicate the ranges (x-axes) of each feature (y-axes). Crosses indicate (arithmetic) means, boxes the ranges (first [Q1] to third [Q3] quartile), with the central line representing the (arithmetic) median (second quartile [Q2]). Whiskers extend to 1.5 times the

interquartile range above Q3 and below Q1. Any data point outside this range is considered an outlier (dots). Mind the different scales for the color codes and SHAP values. Source data are provided as a Source Data file. ChatGPT ADA performed the SHAP analysis on the training sets (number of independent samples: Endocrinologic Oncology dataset, $n = 493$, Gastrointestinal Oncology dataset, $n = 7899$, Otolaryngology dataset, $n = 1209$, and Cardiology dataset, $n = 1712$). Plasma MN plasma concentrations of metanephrine, plasma NMN plasma concentrations of normetanephrine, SDHB succinate dehydrogenase complex iron-sulfur subunit B, Plasma MTY plasma concentrations of methoxytyramine, AGC atypical glandular cells, DNA deoxyribonucleic acid, Chron. chronic, Cong. congenital, Dias. diastolic, Sys. systolic. Note: The feature "Hyp. heart w/ HF & Stg 1–4 Unsp. CKD" refers to "Hypertensive heart with heart failure coexisting with unspecified stage 1–4 chronic kidney disease", while "Prev. hist. PGGLs" refers to "Previous history of Pheochromocytomas and Paragangliomas".

designed our study to include variable data types such as clinical data, demographic data, cytologic data, genetic sequencing data, International Classification of Disease codes, and laboratory values from clinical trials spanning endocrinology, gastrointestinal oncology, genetics, and cardiology. Beyond empowering clinicians to use the clinical data to their patients' advantage, utilizing LLMs for advanced data analysis provides a less costly and more efficient alternative to hand-crafted ML models[32].

In assessing accuracy, validity, and reliability, our study utilized datasets from original studies to gauge the robustness of predictions on previously unseen data. External validation is paramount in evaluating the model and its broader applicability. However, it is worth noting that external validation was absent in the Cardiac Amyloidosis dataset. Across various datasets, models chosen by ChatGPT ADA consistently demonstrated performance on par with, or exceeding, the hand-crafted ML methods used in the original studies. When comparing performance metrics, i.e., AUROC, accuracy, F1-score, sensitivity, and specificity, no significant differences were found between the optimized models re-implemented by our data scientist and those

autonomously chosen by ChatGPT ADA. This observation demonstrates ChatGPT ADA's ability to select, train, and refine suitable and performant ML models autonomously.

We also assessed the risk of overfitting, which occurs when a model excels on training data but may not generalize well to unseen data. When evaluating the model's execution across training and validation datasets for each clinical trial, we observed that ChatGPT ADA had implemented strategies to increase model robustness and generalizability, like regularization, model selection based on validation, and choosing simpler models. However, even though these strategies may be helpful, users should still regularly check performance metrics for signs of overfitting.

Per the design of our prompting strategy, we did not ask ChatGPT ADA for specific explanations of why it selected a particular ML model. However, the tool displayed a surprisingly deep understanding of the clinical trial and appropriate analysis methods. For most clinical-trial datasets, ChatGPT ADA employed a median imputation strategy. In contrast, it used a zero-imputation strategy for the Hereditary Hearing Loss dataset. When asked to explain, ChatGPT ADA indicated that its

choice was informed by the dataset's inherent characteristics and semantics: "The data represents the presence (1) or absence (0) of certain genetic variants in patients. Given this binary representation, median imputation (which would yield either 0 or 1) might introduce bias. In genetic studies, undetected or missing variants are often interpreted as absent, making zero-imputation align with this understanding". This statement is supported by pertinent literature studies[33]. Contrarily, our seasoned data scientist, with a limited command of genetics, used median imputation, underscoring the value of domain-specific knowledge when setting up domain-specific ML methods.

We did not find signs of "hallucinations", i.e., factually erroneous responses[27,34]. Critically, we performed the statistical analysis step-by-step to ascertain the accuracy, reliability, and consistency of the model's outputs. Specific safeguarding measures, such as the provision of intermediary Python code throughout the different phases of building and executing the ML model, have been implemented by the manufacturer and improve comprehensibility and transparency. If coupled with more general safeguarding measures, e.g., increasing user awareness of hallucinations, enhancing the LLM's inherent robustness[35], and implementing regular auditions and quality checks, the tool's validity and reliability can be ascertained[36]. We assessed the consistency in ChatGPT ADA's behavior and analysis choices for each clinical-trial dataset by prompting the tool multiple times in different chat sessions. It consistently selected the same ML model and parameter settings when provided with identical datasets, instructions, and prompts. The only variation occurred when computational resources were limited. In those instances, the tool communicated its primary model choice but temporarily opted for an alternative.

Regarding ease of use, ChatGPT ADA substantially reduces the complexity of developing and implementing ML methods by taking tabular data, suggesting how to deal with it, building the model, and outputting the results in the desired format. Not least due to its ability to communicate with the user, the tool offers a natural and effective way to work with ML models. At the same time, the automatization simplifies the associated workflow. However, as with any innovation, utilizing LLMs in clinical research and practice has multifaceted implications, from data privacy to data security to model interpretability, reliability, and associated ethical concerns[37–41]. Upholding patient data privacy seems particularly challenging as—on the one hand—users may be enticed to disclose confidential (or proprietary) information, let alone sensitive personal data such as race and ethnicity, to use the model most efficiently. On the other hand, OpenAI continuously trains the model using earlier user interactions, including prompts, which are retained as part of the ever-enlarging training data and cannot be deleted. Consequently, it is the user's responsibility to weigh the tool's advantages and benefits against its disadvantages and risks.

ChatGPT ADA, as a tool, democratizes access to advanced ML methods, enabling clinicians and researchers of all backgrounds to harness its capabilities. Besides being a potential cornerstone for their broader utilization in clinical research and practice[42], the improved accessibility holds the potential of (i) accelerating medical research, (ii) confirming or contradicting earlier research, and (iii) improving patient care. However, when using the tool more widely, several potential challenges and limitations must be acknowledged. First, the tool's commercial and proprietary distribution is concerning because (i) its 'black-box' nature limits transparency and may reduce trust in its outputs[43], (ii) commercial bias may be in opposition to the idealized concept of unbiased scientific or clinical deliberation, and (iii) algorithmic bias secondary to the model's potentially skewed foundational data may perpetuate unbalanced outcomes, for example, by not representing those patients adequately that had been underrepresented in the foundational data[44]. In the absence of benchmark publications for comparison, users must be more vigilant in

ascertaining accuracy and reliability, for example, by seeking external validation whenever possible.

Regarding transparency and trust, we conducted a SHapley Additive exPlanations (SHAP) analysis[45] to better understand how ChatGPT ADA works on and with the respective datasets. The tool successfully identified and plausibly quantified the importance of numerous variables across the trials. For instance, its predictions centered on sex, age, and laboratory values (Metastatic Disease [Endocrinologic Oncology] dataset), specific cytologic features such as the presence of atypical glandular cell and nuclear width features (Esophageal Cancer [Gastrointestinal Oncology] dataset), specific gene variants such as c.235delC and p.V37I that are associated with hearing loss[46,47] (Hereditary Hearing Loss [Otolaryngology] dataset), and the previous history of (diagnosed) cardiomyopathy (Cardiac Amyloidosis [Cardiology] Dataset). The in-built ability to autonomously extract key features contributing to the model's predictions increases transparency, improves understanding, and furthers trust in ChatGPT ADA[48].

Our study has limitations: First, clinical ML projects require a reliable and sound database following consistent data pre-processing. While we assessed ChatGPT ADA's performance in the presence of well-curated clinical-trial datasets, real-world clinical data are oftentimes less curated and characterized by data quality issues such as missing and irregular values[49,50]. Successfully applying ML methods to more complex real-world clinical data regularly necessitates more advanced and nuanced pre-processing and statistical methods. Here, ChatGPT ADA's effectiveness remains to be assessed. Second, given their publication in 2021, we cannot exclude the possibility of two original studies[24,25] being part of the training that was concluded in 2021. Given the large sizes of the included datasets consisting of hundreds to thousands of patients, previous publications on the same dataset (or a specific subset), e.g., ref. 51, may have been included as part of the model's training data. Third, even though ChatGPT ADA and the original studies implemented the same model, we found different performance metrics, e.g., for the Cardiac Amyloidosis dataset where random forest classifiers were implemented. Possible sources of variability are the specific approaches used for data pre-processing, dataset splitting, model configurations, and hyperparameter selection. Despite our best efforts to standardize each model's implementation and execution, inter-model comparability is inherently limited. Fourth, because the LLM's response is closely related to how it is prompted[52], it is unclear whether the performance metrics are subject to change if the model is prompted differently. Consequently, our work represents a mere starting point for exploring the potential of LLMs in clinical research and practice. Future research must validate our findings across different medical domains.

In conclusion, advanced LLMs like ChatGPT ADA are a potentially transformative step forward in data-driven medicine, making intricate ML methods more accessible. By way of example, our study demonstrates that such tools may streamline advanced data analyses for researchers, both experienced and inexperienced in ML, and hold the potential to reduce the burden of data pre-processing and model optimization substantially. Given the tools' novelty, limitations, and challenges, they should be applied as enhancements to specialized training, resources, and guidance, not substitutes. Nonetheless, in this dawning era of data-driven medicine, these tools may bridge the chasm between complex ML methods and their practical application in medical research and practice.

## Methods

### Ethics statement

The methods were performed in accordance with relevant guidelines and regulations and approved by the ethical committee of the Medical Faculty of RWTH Aachen University for this retrospective study (Reference No. EK 028/19).

## Patient cohorts

The patient datasets were retrieved from public repositories as indicated in the original studies on metastatic disease prediction[17], esophageal cancer screening[20], hereditary hearing loss[24], and cardiac amyloidosis[25].

In the included Endocrinologic Oncology study[17], cross-sectional data from Germany, Poland, the US, and the Netherlands was used to assess the ability of the dopamine metabolite methoxytyramine to identify metastatic disease in patients with pheochromocytoma or paraganglioma. To this end, ten features were available.

The included Esophageal Cancer study[20] from China was centered on endoscopic screening and included multiple data sources from questionnaires to endoscopy data, i.e., cytologic and epidemiologic data.

The included Hereditary Hearing Loss study[24] contained genetic sequencing data to diagnose this condition in a Chinese cohort. Individuals were categorized based on hearing loss severity and variations in three genes (GJB2, SLC26A4, MT-RNR1).

The included Cardiac Amyloidosis study[25] utilized electronic health records to identify patients with cardiac amyloidosis from a dataset spanning 2008-2019, sourced from IQVIA, Inc., focusing on heart failure and amyloidosis. While the original study used external datasets for validation, these were inaccessible. Therefore, our analysis adhered to the original study's internal validation strategy: 80% as the training set and 20% for testing, resulting in 1712 individuals for training and 430 for testing. For further information on the individual datasets, the reader is referred to Table 1 or the original studies.

## Experimental design

We extracted the original training and test datasets from each clinical trial. All datasets were available in tabular format, albeit in various file formats such as comma-separated values or Excel (Microsoft Corporation). No modifications to the data format, specific data pre-processing or engineering, or selecting a particular ML method were necessary to prompt ChatGPT ADA. GPT-4[13], the current state-of-the-art version of ChatGPT, was accessed online (https://chat.openai.com/) following the activation of the Advanced Data Analysis feature. Initially, we operated the August 3 (2023) version, while, during the project, we transitioned to the September 25 version. A new chat session was started for each trial to exclude memory retention bias.

In the first phase, ChatGPT ADA was sequentially prompted by (i) providing a brief description of the study's background, objectives, and dataset availability, (ii) asking for developing, refining, and executing the optimal ML model based on the individual study's framework and design, and (iii) producing patient-specific predictions (classification probabilities) without revealing the ground truth. The same training and test datasets as in the original studies were used. We deliberately refrained from offering specific ML-related guidance when ChatGPT sought advice on improving prediction accuracy. Instead, ChatGPT ADA was tasked with (i) autonomously choosing the most suitable and precise ML model for the given dataset and (ii) generating predictions for the test data. Figure 2 provides an exemplary interaction with the model.

Using the provided ground-truth test set labels, we calculated the performance metrics for ChatGPT ADA's results using Python (v3.9) using open-source libraries such as NumPy, SciPy, scikit-learn, and pandas.

The performance metrics were compared against those published in the original studies ("benchmark publication"). In some clinical trials, the clinical care specialists' performance was also reported, and these metrics were included for comparison. Notably, inputting and analyzing each dataset through ChatGPT ADA took less than five minutes. Detailed transcripts of the interactions with ChatGPT ADA for every dataset are presented in Supplementary Notes 1–4.

## Data pre-processing and ML model development

In the second phase, a seasoned data scientist re-implemented and optimized the best-performing ML model of the original studies using Python (v3.9) using open-source libraries such as NumPy, SciPy, scikit-learn, and pandas and the same training datasets as outlined above ("benchmark validatory re-implementation"). This re-implementation and optimization was necessary because individual patient predictions were unavailable in the original studies, precluding head-to-head model comparisons and detailed statistical analyses. More specifically, the data scientist optimized the data pre-processing and the ML model in close adherence to the original studies, yet complemented by his expertise and experience while aiming for peak accuracy.

The following provides trial-specific details on the data pre-processing and the conceptualization of the specific ML models.

**Metastatic disease [endocrinologic oncology].** Re-implemented (validatory) ML model: The training set contained 30 missing values, while the test set contained 15 missing values. Median values from the training set were used to impute the missing values in both datasets. Ten distinct feature vectors were constructed from the dataset variables. The feature vectors were partially categorical and partially numerical. The categorical features were: (1) previous history of pheochromocytoma or paraganglioma (yes/no), (2) adrenal/extra-adrenal location of primary tumor (adrenal/extra-adrenal), (3) presence of Succinate Dehydrogenase Complex Iron-Sulfur Subunit B (SDHB) (yes/no/not tested), (4) tumor category of primary tumor (solitary, bilateral, multifocal), and 5) sex (female/male). The numerical features were: (1) age at diagnosis of first tumor [years], (2) spherical volume of primary tumor [$cm^3$], (3) plasma concentration of metanephrine (MN) [pg/ml], (4) plasma concentration of normetanephrine (NMN) [pg/ml], and (5) plasma concentration of methoxytyramine (MTY) [pg/ml]. Categorical data were translated into numerical integer values, e.g., female (0) and male (1) for sex. An Adaptive Boosting (AdaBoost)[18] ensemble tree classifier was employed and optimized using a 10-fold cross-validation grid search. This optimization led to selecting parameters like a maximum depth of 2 for individual decision trees, a count of 200 trees, and a learning rate of 0.01. Stagewise additive modeling was chosen, utilizing a multiclass exponential loss function.

ChatGPT ADA-crafted ML model: A check for missing data mirrored the findings above, leading the model to resort to a median imputation strategy. Numerical data were standardized using standard scaling, while categorical data were converted to integer values. The selected classification technique was a Gradient Boosting Machine (GBM)[19] with parameters set as follows: maximum tree depth: 3, number of trees: 100, minimum samples per leaf: 1, minimum samples for split: 2, and learning rate: 0.1. The logarithmic loss function was the chosen evaluation metric, with the quality of splits being evaluated using the Friedman mean squared error[53]. No validation dataset was incorporated, and the model was not subjected to any specific regularization techniques.

**Esophageal cancer [gastrointestinal oncology].** Re-implemented (validatory) ML model: The training dataset included 147 feature vectors, whereas the test dataset included 169. A comprehensive list of the feature vectors can be found in the literature:[20]. Excess feature vectors in the test set were excluded to maintain consistency, aligning it with the training dataset. Consequently, neither the training nor the test datasets contained missing values. Categorical data were mapped to numerical integer values. Imbalanced dataset distributions were addressed by conferring inverse frequency weights upon the data. In line with the original study, the DS selected the Light Gradient Boosting Machine (LightGBM)[21] with the gradient boosting decision tree algorithm. The configuration for the classifier was as follows: an

unspecified maximum tree depth, 300 trees, a cap of 31 leaves per tree, and a 0.1 learning rate. The logarithmic loss function served as the evaluation metric. The model integrated both $L_1$ and $L_2$ regularization techniques.

ChatGPT ADA-crafted ML model: The pre-processing mirrored the approach above, identifying a class imbalance. The selected classifier was the GBM with parameters including a maximum tree depth of 3, 100 trees, minimum samples per leaf of 1, minimum samples for a split of 2, and a learning rate of 0.1. The model's performance was assessed using the logarithmic loss function, with the quality of tree splits evaluated using the Friedman mean squared error. No validation dataset was incorporated, and the model was not subjected to any specific regularization techniques.

**Hereditary hearing loss [otolaryngology].** Re-implemented (validatory) ML model: The training and test sets included 144 feature vectors, i.e., sequence variants at 144 sites in three genes[24]. The values of the training set were numerical, i.e., 0 (individual has no copies of the altered allele [98.2% of the values]), 1 (individual has one copy of the altered allele [1.6%]), and 2 (individual has two copies of the altered allele [0.2%]), while only one value was missing. The values of the test set were numerical, too, with a similar distribution: 0 (98.3%), 1 (1.5%), and 2 (0.2%), while no values were missing. Missing data points were addressed by imputing the median of the training data. All feature vectors were then subject to MinMax scaling. A Support Vector Machine[23] was the best-performing classifier per the original study, configured with the Radial Basis Function kernel, gamma set to 1, and enabled shrinking. Model optimization leveraged a 5-fold stratified cross-validation using grid search. The regularization cost parameter was defined at 100.

ChatGPT ADA-crafted ML model: The pre-processing was closely aligned with the methodology above, with one notable exception: Missing data was addressed by zero-imputation. The classifier chosen was the Random Forest (RF)[22], with the following framework parameters: no explicitly defined maximum depth for individual trees, tree count of 100, minimum samples per leaf of 1, and minimum samples per split of 2. At each split, the features considered were the square root of the total features available. 5-fold cross-validation was employed without the use of a grid search. Regularization was achieved by averaging predictions across multiple trees. Bootstrapping was chosen to create diverse datasets for training each decision tree in the forest.

**Cardiac amyloidosis [cardiology].** Re-implemented (validatory) ML model: The dataset comprised 1874 numerical (0 or 1, indicating the presence or absence) feature vectors[25]. There was no value missing in the dataset. The feature vectors underwent standard scaling for normalization. The classifier chosen was the RF, with the following parameters: maximum depth for individual trees of 20, total tree number of 200, minimum samples per leaf of 2, and minimum samples per split of 5. For each tree split, the square root of the total features determined the number of features considered. A 5-fold cross-validation was combined with a grid search for optimization. Regularization was effectuated by averaging the predictions over multiple trees. The model did not utilize bootstrapping.

ChatGPT ADA-crafted ML model: As there was no missing value in the dataset and the values were binary, the data underwent no scaling or standardization. The selected classifier was the RF. Parameters for the model were as follows: an unspecified maximum depth for individual trees, a tree count of 1000, minimum samples per leaf of 1, and minimum samples per split of 2. For each tree split, the features considered were the square root of the total feature count. The model was validated using 5-fold cross-validation without grid search. Regularization was achieved by averaging predictions across several trees, and the model utilized bootstrapping[22,54].

Because ChatGPT ADA provides all intermediary Python code during data pre-processing and ML model development and execution, we meticulously analyzed the code for accuracy, consistency, and validity.

## Explainability analysis
We used SHapley Additive exPlanations (SHAP)[26] to analyze feature contributions to the model's predictions. ChatGPT ADA was tasked with autonomously performing a SHAP analysis to be narrowed down to the top 10 features. To ensure accuracy, the seasoned data scientist (S.T.A. with five years of experience) reviewed the Python code provided by ChatGPT ADA and re-implemented the procedure in Python using SHAP library[26] with TreeExplainer[55] to confirm the model's outputs.

## Reproducibility analysis
We evaluated the consistency of the tool's responses using separate chat sessions (to avoid memory retention bias), yet the same datasets, instructions, and prompts on three consecutive days. The model consistently reported the same responses and qualitative and quantitative findings.

## Statistical analysis and performance evaluation
The quantitative performance evaluation was performed using Python (v3.9) and its open-source libraries, such as NumPy and SciPy. Unless noted otherwise, performance metrics are presented as mean, standard deviation, and 95% confidence interval (CI) values.

Using the published ground-truth labels from the original studies as reference ("benchmark publication"), we calculated a range of performance metrics based on ChatGPT ADA's predictions of the test set labels: AUROC, accuracy, F1-score, sensitivity, and specificity. These performance metrics are presented alongside those reported in the original studies, if available (Table 2).

Once the per-patient predictions were available following the re-implementation and optimization of the select ML models ("benchmark validatory re-implementation"), we calculated the performance metrics outlined above using the ground-truth labels for the re-implemented (validatory) ML models and their ChatGPT ADA-based counterparts. We adopted bootstrapping[54] with replacements and 1000 redraws on the test sets to ascertain the statistical spread (in terms of means, standard deviations, and 95% confidence intervals), and to determine if the metrics were significantly different. We adjusted for multiple comparisons based on the false discovery rate, setting the family-wise alpha threshold at 0.05. Notably, the comparative evaluation of the performance metrics was conducted in a paired manner. Bootstrapping was applied to both models. The threshold for calculating the F1-score, sensitivity, and specificity was chosen based on Youden's criterion[56].

## Reporting summary
Further information on research design is available in the Nature Portfolio Reporting Summary linked to this article.

## Data availability
The datasets utilized in this study were extracted from public repositories. The raw data for predicting metastatic disease in pheochromocytoma or paraganglioma[17] is available on *Zenodo*: https://doi.org/10.5281/zenodo.7749613. Esophageal cancer screening-trial data[20] is available on GitHub: https://github.com/Gaoоoooye/Esophageal-cAncer-Screening-Trial. The hereditary hearing loss trial data[24] is available on Mendeley: https://data.mendeley.com/datasets/6mh8mpnbgv/1. The data on cardiac amyloidosis ("derivation dataset") is available per the original study[25] at https://www.nature.com/articles/s41467-021-22876-9. Source data of Figures are provided in this paper. Source data are provided in this paper.

## Code availability

ChatGPT Advanced Data Analysis (previously known as "Code Interpreter") was utilized for analyses and may be accessed via https://chat.openai.com for ChatGPT Plus users. Source codes for training, evaluating, and optimizing the ML models, as well as for data pre-processing, statistical analysis, and visualizations, are publicly available at https://github.com/tayebiarasteh/LLMmed. The code was developed in Python v3.9.18 using open-source libraries including shap (v0.44.0), NumPy (v1.26.2), SciPy (v1.11.4), lightgbm (v4.1.0), mne (v1.6.0), pandas (v2.1.1), and scikit-learn (v1.3.0). All source codes are permanently archived on *Zenodo* and are accessible via ref. 57.

## Materials availability

The hardware used in our experiments included an Intel CPU with eight cores and 16 GB RAM. No GPU was utilized.

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

## Acknowledgements

S.T.A. is funded and partially supported by the Radiological Cooperative Network (RACOON) under the German Federal Ministry of Education and Research (BMBF) grant number 01KX2021. S.N. and D.T. were supported by grants from the Deutsche Forschungsgemeinschaft (DFG) (NE 2136/3-1, TR 1700/7-1). D.T. is supported by the German Federal Ministry of Education (TRANSFORM LIVER, 031L0312A; SWAG, 01KD2215B) and the European Union's Horizon Europe and Innovation program (ODELIA [Open Consortium for Decentralized Medical Artificial Intelligence], 101057091).

## Author contributions

S.T.A., D.T., and S.N. designed the study and performed the formal analysis. The manuscript was written by S.T.A. and corrected by D.T. and S.N. The experiments were performed by S.T.A. The statistical analysis was performed by S.T.A. and M.L. The illustrations were designed by S.T.A., M.L., D.T., and S.N. The literature research was conducted by S.T.A., D.T., and S.N. J.N.K., C.K., D.T., and S.N. provided clinical expertise. S.T.A., T.H., M.L., J.N.K., and D.T. provided technical expertise. All authors read the manuscript and agreed to the submission of this paper.

## Funding

## Competing interests

J.N.K. declares consulting services for Owkin, France; DoMore Diagnostics, Norway, and Panakeia, UK. Furthermore, J.N.K. holds shares in StratifAI GmbH and has received honoraria for lectures by Bayer, Eisai, MSD, BMS, Roche, Pfizer, and Fresenius. D.T. holds shares in StraifAI GmbH, Germany, and received honoraria for lectures by Bayer. The other authors declare no competing interests.
