## [Peer Review File · Nature Communications]

Reviewers' Comments:

Reviewer #1:

Remarks to the Author:

I just have one minor comment: in the discussion, regarding the SHAP explainability analysis, can you please briefly contextualise the identified features? You can do so by comparing the identified features to the original papers and by looking at domain-specific literature. For example, is there evidence in the literature corroborating that variant c.235delC is associated to hearing loss? etc.

Reviewer #2:

Remarks to the Author:

The study tests ChatGPT's Advanced Data Analytics (ADA) "AutoML" capabilities to develop and evaluate common machine learning models for different clinical prediction tasks based on 4 example clinical (trial) datasets.

With a focus on the previous Reviewer #2 comments, it appears that this revision largely addresses those comments adequately. Most of the prior reviewer comments were asking for additional discussion and clarifications on different methods used and implications, which the authors have respectively responded to and expounded upon in the main text.

This included a more thorough description of model hyperparameter selection, SHAP explainability assessments, bootstrapped confidence intervals to assess metric uncertainty.

Comments and Suggestions below are minor, largely revolving around clarity and optimizing formatting.

It was not entirely clear if the revision addressed the prior reviewer comments about whether multiple repeat prompts were attempted to assess for consistency in ChatGPT ADA's behavior and choices of analysis.

The revision could benefit from a sharper description of the previous reviewer's comments between the well-curated (trial) datasets used for testing compared to the real-world clinical data sources that are likely to have much more missingness and irregularities. Half the battle in clinical machine learning projects is typically the data pre-processing to get heterogeneous clinical data in irregular data formats into a well-structured data frame to enable the AutoML process.

As a more general presentation comment, many of the chat interaction screenshots (e.g. Figure 2) appear illegible at lower resolution when formatted into PDF.

These need to be high resolution in publication, or better yet, consider simply including the text of the chat logs (as in the Supplementary material chat logs).

These are essential to the value of this manuscript, as they provide the blueprint for readers to replicate the process in their own studies.

Text format would be more useful than screenshots, as it would facilitate copy-paste for readers to attempt their own instruction prompts.

Reviewer #3:
None

Point-by-Point Responses to the Editor's and the Reviewers' Comments

Title:

Large language models streamline automated machine learning for clinical studies

Reference number:

NCOMMS-23-48343-T

Journal:

General Reply:

We thank the Editor and the Reviewers for thoroughly reviewing our manuscript. We hope to have satisfactorily addressed all remaining issues. After our manuscript's thorough revision, we would be delighted to see it published in *Nature Communications*.

Please find our responses to the Reviewers' comments below, along with the resulting changes to the manuscript.

In the following, we have addressed the comments point by point. Please note that all changes made to the original manuscript have been highlighted in yellow.

Reply to Reviewer #1

NCOMMS-23-48343-T

Title:

Large language models streamline automated machine learning for clinical studies

General Comment: *“I just have one minor comment: in the discussion, regarding the SHAP explainability analysis, can you please briefly contextualise the identified features? You can do so by comparing the identified features to the original papers and by looking at domain-specific literature. For example, is there evidence in the literature corroborating that variant c.235delC is associated to hearing loss? etc.”*

Authors' Response and Action: We appreciate the Reviewer's suggestion to provide more context for the features identified in the SHAP explainability analysis. Consequently, we have consulted relevant domain-specific literature to corroborate the association of specific genetic variants with clinical conditions.

Indeed, as noted in the literature [1], the variant c.235delC is associated with hearing loss. Additionally, we have included another variant, p.V37I, which is also linked to hearing loss [2] and exhibited the highest impact after c.235delC according to the SHAP analysis results (**Fig. 4** [manuscript]).

Accordingly, we have updated the relevant section of the **Discussion**:

“Regarding transparency and trust, we conducted a SHapley Additive exPlanations (SHAP) analysis⁴⁵ to better understand how ChatGPT ADA works on and with the respective datasets. The tool successfully identified and plausibly quantified the importance of numerous variables across the trials. For instance, its predictions centered on sex, age, and laboratory values (Metastatic Disease [Endocrinologic Oncology] dataset), specific cytologic features such as the presence of atypical glandular cell and nuclear width features (Oesophageal Cancer [Gastrointestinal Oncology] dataset), specific gene variants such as c.235delC and p.V37I that are associated with hearing loss^{46,47} (Hereditary Hearing Loss [Otolaryngology] dataset), and the previous history of (diagnosed) cardiomyopathy (Cardiac Amyloidosis [Cardiology] Dataset). The in-built ability to autonomously extract key features contributing to the model's predictions increases transparency, improves understanding, and furthers trust in ChatGPT ADA⁴⁸.”

(page 8)

- [1] Xia, Hong, et al. "GJB2 c. 235delC variant associated with autosomal recessive nonsyndromic hearing loss and auditory neuropathy spectrum disorder." *Genetics and Molecular Biology* 42 (2019): 48-51.
- [2] Shen, Na, et al. "Association between the p. V37I variant of GJB2 and hearing loss: a pedigree and meta-analysis." *Oncotarget* 8.28 (2017): 46681.

Reply to Reviewer #2

NCOMMS-23-48343-T

Title:

Large language models streamline automated machine learning for clinical studies

General Comment: *"The study tests ChatGPT's Advanced Data Analytics (ADA) "AutoML" capabilities to develop and evaluate common machine learning models for different clinical prediction tasks based on 4 example clinical (trial) datasets.*

With a focus on the previous Reviewer #2 comments, it appears that this revision largely addresses those comments adequately. Most of the prior reviewer comments were asking for additional discussion and clarifications on different methods used and implications, which the authors have respectively responded to and expounded upon in the main text.

This included a more thorough description of model hyperparameter selection, SHAP explainability assessments, bootstrapped confidence intervals to assess metric uncertainty.

Comments and Suggestions below are minor, largely revolving around clarity and optimizing formatting."

Authors' Response: We thank the Reviewer for their constructive feedback and are pleased to hear that our revisions largely addressed the previous concerns. We are committed to implementing the suggested minor revisions for enhanced clarity and formatting.

Comment 1: *" It was not entirely clear if the revision addressed the prior reviewer comments about whether multiple repeat prompts were attempted to assess for consistency in ChatGPT ADA's behavior and choices of analysis."*

Authors' Response and Action: We thank the Reviewer for highlighting the critical aspect of consistency in ChatGPT ADA's responses to repeat prompts. In our previous response (to the first Reviewer #2), we confirmed that multiple repeat prompts were utilized to assess the consistency of ChatGPT ADA's behavior and choice of analysis. For each clinical trial dataset, we prompted ChatGPT ADA in a standardized manner and multiple times across different chat sessions. After registering its responses, we meticulously studied the suggested methodologic approach and Python code lines. Throughout these tests, we did not observe signs of hallucinations (i.e., fabrications) or random model selection. ChatGPT ADA consistently chose the same ML model and

parameter settings given the same datasets, instructions, and prompts. The sole exception occurred when computational resources were constrained. In these instances, ChatGPT ADA indicated its primary model choice and temporarily shifted to an alternate option due to resource constraints.

In reflection of the Reviewer's comment, we have revised the respective section of the **Discussion** section for better clarity:

"We did not find signs of "hallucinations", i.e., factually erroneous responses^{27,34}. Critically, we performed the statistical analysis step-by-step to ascertain the accuracy, reliability, and consistency of the model's outputs. Specific safeguarding measures, such as the provision of intermediary Python code throughout the different phases of building and executing the ML model, have been implemented by the manufacturer and improve comprehensibility and transparency. If coupled with more general safeguarding measures, e.g., increasing user awareness of hallucinations, enhancing the LLM's inherent robustness³⁵, and implementing regular auditions and quality checks, the tool's validity and reliability can be ascertained³⁶. We assessed the consistency in ChatGPT ADA's behavior and analysis choices for each clinical trial dataset by prompting the tool multiple times in different chat sessions. It consistently selected the same ML model and parameter settings when provided with identical datasets, instructions, and prompts. The only variation occurred when computational resources were limited. In those instances, the tool communicated its primary model choice but temporarily opted for an alternative."

(pages 6 and 7)

Comment 2: " The revision could benefit from a sharper description of the previous reviewer's comments between the well-curated (trial) datasets used for testing compared to the real-world clinical data sources that are likely to have much more missingness and irregularities. Half the battle in clinical machine learning projects is typically the data pre-processing to get heterogeneous clinical data in irregular data formats into a well-structured data frame to enable the AutoML process."

Authors' Response and Action: This equally important and insightful comment touches on the frequently unglamorous realities of data-driven clinical research. Undoubtedly, a gap exists between (well-curated) trial datasets and (ill-curated) real-world clinical datasets. Acknowledging the importance of this aspect, we have revised the **Limitations** of our **Discussion** accordingly:

"First, clinical ML projects require a reliable and sound database following consistent data pre-processing. While we assessed ChatGPT ADA's performance in the presence of well-curated clinical trial datasets, real-world clinical data are oftentimes less curated and characterized by data quality issues such as missing and irregular values^{49,50}.

Successfully applying ML methods to more complex real-world clinical data regularly necessitates more advanced and nuanced pre-processing and statistical methods. Here, ChatGPT ADA's effectiveness remains to be assessed. [...]"
(page 8)

Comment 3: *"As a more general presentation comment, many of the chat interaction screenshots (e.g. Figure 2) appear illegible at lower resolution when formatted into PDF. These need to be high resolution in publication, or better yet, consider simply including the text of the chat logs (as in the Supplementary material chat logs). These are essential to the value of this manuscript, as they provide the blueprint for readers to replicate the process in their own studies. Text format would be more useful than screenshots, as it would facilitate copy-paste for readers to attempt their on instruction prompts."*

Authors' Response and Action: We thank the Reviewer for the valuable feedback regarding the presentation of our chat interactions. We understand the importance of clarity and accessibility of these illustrations. To address this, we have converted our interactions with the tool, i.e., all prompts and responses, to text format and made them available as **Supplementary Notes 1-4**. This format facilitates replication by others.

Regarding **Fig. 2**, we intend to visually demonstrate the interaction with ChatGPT ADA. Based on the Reviewer's comments, we have revised this figure to include only key parts of the conversation, which allows for higher resolution and better legibility. Additionally, we have decided to remove **Supplementary Fig. 1**, aligning with our commitment to maintain high-quality visual aids throughout the manuscript and considering the redundancy with the revised **Fig. 2**.